# High-Throughput MicroRNA Profiling of Vitreoretinal Lymphoma: Vitreous and Serum MicroRNA Profiles Distinct from Uveitis

**DOI:** 10.3390/jcm9061844

**Published:** 2020-06-12

**Authors:** Teruumi Minezaki, Yoshihiko Usui, Masaki Asakage, Masakatsu Takanashi, Hiroyuki Shimizu, Naoya Nezu, Akitomo Narimatsu, Kinya Tsubota, Kazuhiko Umazume, Naoyuki Yamakawa, Masahiko Kuroda, Hiroshi Goto

**Affiliations:** 1Department of Ophthalmology, Tokyo Medical University, 6-7-1 Nishi-shinjuku, Shinjuku-ku, Tokyo 160-0023, Japan; terumi2098@gmail.com (T.M.); patty.m.best@gmail.com (M.A.); sardine_harbor@yahoo.co.jp (H.S.); naoya.nezu@gmail.com (N.N.); a.narimatsu1128@gmail.com (A.N.); kinyatomoko1018@yahoo.co.jp (K.T.); kazuhiko-uma@kvf.biglobe.ne.jp (K.U.); yamakawa@tokyo-med.ac.jp (N.Y.); goto1115@tokyo-med.ac.jp (H.G.); 2Department of Molecular Pathology, Tokyo Medical University, 6-7-1 Nishi-shinjuku, Shinjuku-ku, Tokyo 160-0023, Japan; m-takana@tokyo-med.ac.jp (M.T.); kuroda@tokyo-med.ac.jp (M.K.)

**Keywords:** vitreoretinal lymphoma, microRNA, uveitis, machine learning

## Abstract

Purpose: Vitreoretinal lymphoma (VRL) is a non-Hodgkin lymphoma of the diffuse large B cell type (DLBCL), which is an aggressive cancer causing central nervous system related mortality. The pathogenesis of VRL is largely unknown. The role of microRNAs (miRNAs) has recently acquired remarkable importance in the pathogenesis of many diseases including cancers. Furthermore, miRNAs have shown promise as diagnostic and prognostic markers of cancers. In this study, we aimed to identify differentially expressed miRNAs and pathways in the vitreous and serum of patients with VRL and to investigate the pathogenesis of the disease. Materials and Methods: Vitreous and serum samples were obtained from 14 patients with VRL and from controls comprising 40 patients with uveitis, 12 with macular hole, 14 with epiretinal membrane, 12 healthy individuals. The expression levels of 2565 miRNAs in serum and vitreous samples were analyzed. Results: Expression of the miRNAs correlated significantly with the extracellular matrix (ECM) ‒receptor interaction pathway in VRL. Analyses showed that miR-326 was a key driver of B-cell proliferation, and miR-6513-3p could discriminate VRL from uveitis. MiR-1236-3p correlated with vitreous interleukin (IL)-10 concentrations. Machine learning analysis identified miR-361-3p expression as a discriminator between VRL and uveitis. Conclusions: Our findings demonstrate that aberrant microRNA expression in VRL may affect the expression of genes in a variety of cancer-related pathways. The altered serum miRNAs may discriminate VRL from uveitis, and serum miR-6513-3p has the potential to serve as an auxiliary tool for the diagnosis of VRL.

## 1. Introduction

Vitreoretinal lymphoma (VRL) is a rare malignancy arising in the vitreous and subretinal space. This disease is characterized by high rate of relapse and may manifest before, after, or simultaneously with central nervous spread, which is generally incurable due to a lack of active systemic therapies. In Japan, multicenter epidemiologic surveys of uveitis were conducted in 2002 [1] and 2009 [2]. According to these surveys, the prevalence of VRL is increasing because of the advances in diagnostic tests and the increase in recognition of the disease. However, VRL remains a masquerade syndrome and is frequently misdiagnosed as chronic idiopathic uveitis due to the nonspecific clinical presentations and lack of definitive biomarker [3,4]. The gold standard tests for the diagnosis of VRL are cytology, polymerase chain reaction (PCR) analysis for monoclonality of immunoglobulin H (IgH), and interleukin (IL)-10 and IL-6 cytokine levels. Cytology had a low detection rate (44.5%) in our Japanese survey [5]. PCR analysis for IgH receptor rearrangements also remains limited, with suboptimal detection rates (80% in VRL with vitreous opacity and 75% in VRL with subretinal infiltration) [5]. IL-10 is the established diagnostic biomarker for VRL (90% detection in cases with vitreous opacity and subretinal infiltration), despite the limitation that elevated vitreous IL-10 level is not diagnostic of intraocular or central nervous system lymphoma [6]. On the other hand, the IL-10/IL-6 ratio is a useful diagnostic tool: a ratio higher than one has been reported to indicate VRL and a ratio lower than one to indicate uveitis [7]. However, our group has found that 10% to 20% of VRL cases have IL-10/IL-6 ratio less than 1 [5]. Recently, several groups have reported a new PCR test for myeloid differentiation primary response (MYD)^L265P^ mutations as a new diagnostic tool for VRL with 70% to 89% sensitivity [8,9]. In recent decades, there has been little improvement in the survival rate of central nervous system lymphoma developed from VRL, with five-year survival rates from below 59% to 69% [5,10,11]. Around 50% to 90% of VRL patients present with central nervous system involvement [5,12,13]. Clinical outcomes of VRL differ significantly depending on clinical and prognostic variables. Further elucidation of the underlying molecular mechanism and identification of reliable diagnostic biomarkers are required.

Mammalian microRNAs (miRNAs) are small non-coding regulatory RNAs (18–25 nucleotides long) that have the function of suppressing gene translation and degrading mRNA by binding to the sequences in the target mRNAs [14]. Thus, miRNAs function by negatively regulating gene expression at mRNA and protein levels. They play important roles as fundamental regulators of various physiological functions, and have been shown to be particularly associated with cancer. More than 50% of the miRNA genes are located in tumor-related genomic regions or fragile sites [15]. Thus, miRNA studies may lead to better understanding of disease mechanisms and may uncover new diagnostic markers for classification of disease subgroups and analysis of severity in complex diseases. The concept that miRNAs participate in the pathogenesis of VRL, especially refractory diseases with unidentified mechanisms, may lead to development of novel effective treatment. There were only 2 studies investigating human miRNA directly from vitreous samples. Kakkassery et al. [16] studied the expression of three known central nervous system lymphoma‒related miRNAs (miR-92, miR-19b and miR-21) in vitreous samples from VRL patients using vitritis samples as control. Tuo et al. [17] also reported that, among 168 miRNAs in the test array, only miR-155 expression in vitreous samples was significantly lower in VRL patients compared with uveitis patients as control. High-throughput miRNA sequencing, currently capable of analyzing more than 2500 miRNAs, has received tremendous attraction as a tool to identify miRNAs specifically associated with diseases. However, global miRNA profiling in vitreous and serum from patients with VRL using uveitis with vitreous opacity as control has not been conducted.

We hypothesized that miRNA profiling can be used for differential diagnosis of VRL and can improve understanding of the pathogenesis of VRL. In the present study, we performed a comprehensive miRNA microarray analysis on serum and vitreous samples from VRL patients manifesting vitreous opacity as well as on samples from control patients with uveitis, macular hole and epiretinal membrane, aiming to elucidate the pathophysiology of VRL as well as to identify new diagnostic biomarker or therapeutic targets. To the best of our knowledge, such a study has not been reported in the literature. We identified specific miRNA signatures in vitreous and serum associated with VRL, which discriminate VRL from uveitis with vitreous opacity and other controls. These deregulated miRNAs target signaling pathways typically implicated in VRL pathogenesis, such as programmed cell death (PD)-1/PD-L1, WNT/β-catenin, B-cell receptor signaling and nuclear factor-kappa B (NF-κβ) pathway [18,19,20]. Our findings may propose potential biomarkers and a framework for VRL pathogenesis, contribute to the development of new diagnostic biomarkers and guide the next steps of clinical investigations.

## 2. Materials and Methods

Fourteen patients with VRL diagnosed and treated at Tokyo Medical University Hospital in 2017–2018 were studied. The VRL patients comprised five males and nine females, aged 66.6 ± 12.0 years. Excluded from the study were patients with preoperative trauma, pre-existing macular pathologies (such as age-related macular degeneration), vitreous hemorrhage, immunodeficiency, and diabetes mellitus, all of which are likely to influence the vitreous and serum expression levels of miRNAs. The following clinical data were extracted from each patient: sex, age at diagnosis of VRL, ocular involvement, main ocular lesions at initial diagnosis, primary organ involved, presence of CNS central nervous system involvement, pattern of spread, cytopathology, presence of IgH chain gene rearrangement, relapse after first diagnosis of vitreoretinal B-cell lymphoma, and current status (Table 1). Both vitreous and serum samples were available from five VRL patients and nine uveitis patients.

The patient control group consisted of 40 patients having uveitis with vitreous opacity (eight males and 32 females; aged 68.1 ± 14.1 years; 20 patients with ocular sarcoidosis and 20 with uveitis of unknown etiology) as well as 12 patients with macular hole and 14 with epiretinal membrane (12 males and 14 females; aged 68.7 ± 9.5 years) and 12 healthy individuals (six males and six females; aged 53.9 ± 20.5 years). None of the control patients had associated vitreoretinopathy. VRL patients were divided into two groups according to clinical features: diffuse vitreous opacities with or without multiple subretinal white lesions. Demographic, clinical and laboratory data (including white blood cell count and soluble IL-2 receptor level) of VRL patients and controls are shown in Table 1.

VRL patients were age-matched with patients with uveitis (age comparison, *p* = 0.38), patients with macular hole/epiretinal membrane (*p* = 0.56), and healthy individuals (*p* = 0.40). All patients were Asian adults. All patients underwent brain magnetic resonance imaging (MRI). Vitreous samples were harvested from the mid-vitreous region at the start of a standard 3-port 25-gauge vitrectomy. The vitreous was removed by a vitreous cutter before intraocular infusion. Then, a complete vitrectomy with infusion of balanced salt solution was performed. Undiluted and diluted vitreous specimens were delivered immediately to the Cytology and Molecular Laboratory (performed at SRL, Tokyo, Japan). The samples were stored immediately at −80 °C until assay.

The diagnosis of VRL was based on clinical features and results of cytology, cytokine analysis, and PCR test using vitreous sample after diagnostic vitrectomy. Cytopathologic and immunocytochemical evaluations were performed on undiluted vitreous samples to evaluate features of the lymphoma, as described previously [21]. The samples were diagnosed without knowledge of the findings of IgH chain gene rearrangement or concentrations of IL-10 and IL-6. Undiluted vitreous specimens were also used in analysis of miRNA. Diluted vitreous samples were used in PCR for analysis of IgH chain gene rearrangement to detect B-cell clonal expansion (performed at SRL, Tokyo, Japan).

Written informed consent was obtained from all the participants in the study. The study was approved by the Ethical Committee of the Tokyo Medical University Hospital, Tokyo, Japan (2016-162). All investigations were conducted according to the principles of the Helsinki declaration.

### 2.1. Blood Sample Collection

From each participant enrolled in this study, a sample of venous blood (approximately 5.0 mL) was collected in BD vacutainer tube using 21-gauge needle, then centrifuged (1000× *g*, room temperature, 15 min) to collect serum, which was stored at −80°C until assay.

### 2.2. Microarray Analysis

Total RNA was extracted from serum and vitreous samples using miRNeasy Mini Kit (Qiagen GmbH, Hilden, Germany) following the manufacturer’s instructions. Gene tip miRNA was extracted from the serum and vitreous samples using 3D-Gene^®^RNA extraction reagent from a liquid sample kit (Toray Industries, Inc., Kamakura, Kanagawa, Japan) and concentrated. Fluorescent labeling of RNA was performed using 3D-Gene^®^ miRNA Labeling kit. Labeled RNA was hybridized to a 3D-Gene^®^ Human miRNA Oligo Chip (Toray Industries, Inc.) designed to detect 2565 mature human miRNA sequences registered in miRBase release 21 (http://www.mirbase.org/). The chip was scanned using a 3D-Gene^®^ Scanner, miRNAs with signals higher than the background signal were selected (positive call), and only miRNAs with positive call were used in subsequent analyses. The miRNA signal values were standardized by global normalization (log conversion of data and median alignment) [22].

### 2.3. Bioinformatic Analysis and Statistical Analysis

Relative expression level of each miRNA was validated using one-way analysis of variance or t-test (*p* < 0.05). MicroRNAs with expression levels of at least 2-fold difference (|log_2_ fold change| > 1 and *p* < 0.05) in the test sample versus control sample were analyzed.

Principal component analysis was used to discriminate between biological samples based on the distances of a reduced set of new variables (principal components), using the top three principal components to depict the results in three dimensions. Unsupervised hierarchical clustering analysis was performed using an algorithm based on Pearson correlation and the average linkage method. Differentially expressed genes and miRNAs in any two groups were identified using criteria comprising *p* value and fold change. Genes targeted by significantly modulated miRNAs were identified using miRDB database, an integrative database for prediction of human functional microRNA targets [23]. Pathway enrichment analyses of miRNA gene targets and differentially expressed genes in VRL were performed using DIANA-mirPath v.3.0 [24]. The web-based computational tools DIANA-mirPath v.3.0 and DAVID 6.8 (https://david.ncifcrf.gov/) were used to predict the target genes and altered pathways of the differentially expressed miRNAs. Cytoscape 3.7.1 (http://manual.cytoscape.org/en/stable/) was used to create plots of the relations between Extracellular matrix (ECM)-receptor interaction and microRNA, and between IL-10 and microRNA.

Statistical analyses were performed using JMP version 13 (Business Unit of SAS, Cary, NC, USA). Continuous variables were compared between two groups using Student’s t-test. Differences were considered significant at *p* values less than 0.05. Machine learning algorithms were implemented by random forest algorithm using R environment (https://cran.r-project.org/).

## 3. Results

### 3.1. High-Throughput Vitreous and Serum miRNA Profiling in VRL Patients

Since a global miRNA expression analysis with updated coverage of miRNA sequences for VRL has not been reported, we provide detailed descriptions of miRNAs associated with VRL by investigating the transcription of a large number of miRNA sequences in vitreous and serum of VRL patients by microarray strategy. Vitreous samples obtained from six patients (eight eyes) with VRL and 26 age- and sex-matched macular hole/epiretinal membrane and 16 uveitis patients were analyzed using dedicated and high-density array with a coverage of more than 2500 human miRNA transcripts and all mature miRNA sequences registered in miRBase release 21.

### 3.2. Vitreous miRNAs

Microarray analysis revealed a large number of modulated miRNAs in vitreous samples, which satisfied the t-test *p* value criterion (*p* < 0.05) and the fold change criterion (>2), showing a robust and statistically significant differentiation between VRL and macular hole/epiretinal membrane. Such a large number of modulated transcripts clearly reflect the high performance of the array in the detection of a wide range of miRNA sequences. We thereafter refined our analysis by selecting only modulated miRNAs annotated as having high confidence in miRDB database (http://www.mirdb.org/), which would render our results more informative. Using these criteria, we selected 1771 modulated miRNAs as shown in Appendix A.

Interestingly, 1714 (control 1436 + uveitis 278) miRNAs were upregulated and only 57 (control 42 + uveitis 15) were downregulated in the vitreous (Table 2).

Hierarchical cluster analysis, an unsupervised approach, was then performed to investigate miRNA variations between VRL patients and macular hole/epiretinal membrane patients (Figure 1a,b).

As shown in Figure 1a, there was a significant separation between the vitreous samples of VRL patients and those of macular hole/epiretinal membrane patients. A miRNA pathway analysis of these vitreous miRNAs suggested that the Hippo signaling pathway was the most relevant among the pathways enriched in VRL-modulated miRNA target genes obtained by gene union analysis (Figure 2).

Further pathway union analysis showed that the ECM-receptor interaction pathway was the most significantly influenced pathway (*p* < 0.05), followed by prion diseases, glycosphingolipid biosynthesis - lacto and neolacto series, TGF-beta signaling pathway, and Hippo signaling pathway (Appendix A).

The results suggest that VRL patients have a significantly distinct vitreous miRNA profile compared to macular hole/epiretinal membrane patients. Further analysis revealed that ten miRNAs (miR-1273d, miR-133b, miR-146a-5p, miR-181-5p, miR-193b-3p, miR-221-3p, miR-326, miR-345-5p, miR-422a, miR-4655-3p) were commonly up-regulated in the vitreous and serum (Appendix A); among them, miR-326 had the largest area under the receiver operating characteristic (ROC) curve, with a value of 0.98.

### 3.3. Serum miRNAs

Likewise, we used the same methodology to perform serum miRNA profiling in patients with VRL and healthy individuals. The serum samples of 13 patients with VRL, 12 healthy controls and 34 uveitis patients were tested. A statistical analysis showed that 36 serum miRNAs were significantly upregulated, while 49 serum miRNAs were significantly downregulated. A hierarchical cluster analysis of miRNAs showed that there was a separation to some extent between serum samples of patients with VRL and those of controls (Figure 3).

A miRNA pathway analysis of these miRNAs suggested that fatty acid biosynthesis was the most relevant pathway among the pathways enriched in VRL-modulated miRNA targets genes obtained by gene union analysis (Figure 4).

Further pathway union analysis showed that the pathway of morphine addiction was the most significantly influenced pathway (*p* < 0.05), followed by fatty acid biosynthesis, thyroid hormone synthesis, mucin type O-glycan biosynthesis, and Hippo signaling pathway (Appendix A).

These results indicate that, even in serum, patients with VRL had a significantly distinct miRNA profile.

### 3.4. Comparison Between Vitreous and Serum miRNA Profiles of VRL Versus Uveitis

To verify whether serum or vitreous miRNA profile can discriminate VRL from uveitis with vitreous opacity, we used the same methodology for vitreous and serum miRNA profiling in VRL and compared with the results in uveitis with vitreous opacity. A statistical analysis showed that 278 vitreous miRNAs and 15 serum miRNAs were upregulated, while 15 vitreous miRNAs and one serum miRNA were downregulated in VRL compared to uveitis with vitreous opacity (Table 3).

A hierarchical cluster analysis of miRNAs showed that there was a separation to some extent between vitreous and serum samples of patients with VRL and those of patients with uveitis (Figure 5).

MiR-6513-3p, 138-2f-3p and 4445-3p were upregulated commonly in vitreous and serum, and miR-6513-3p had the largest area under the ROC curve of the three, with a value of 0.84. These results indicate that miR-6513-3p may be a potential diagnostic biomarker of VRL, which is important for differentiation of VRL from uveitis with vitreous opacity (Figure 6).

### 3.5. Pathway Enrichment Analysis of miRNAs Deregulated in VRL

In the second part of our analysis, we aimed to identify all the molecular pathways that were targeted by the selected miRNA by performing pathway enrichment analyses based on the annotated gene targets registered in DIANA-mirPath. Detailed analysis of the most relevant pathways of VRL was done by performing a search using the KEGG database for potential compound identities and relevant pathways. The software allows evaluation of the miRNA regulatory effects and identification of regulatory pathways based on predicted and validated miRNA‒target interactions. The predominant pathways were involved in ECM-receptor interaction (Figure 7).

Interestingly, according to pathway analysis using DIANA-mirPath based on combinations of differentially expressed ECM‒receptor interaction pathways, a well described pathway for breast cancer [25] was suggested to play a central role in molecular pathology of VRL when compared to uveitis with vitreous opacity. Figure 8 shows the predicted genes targeted by miRNAs in the ECM-receptor interaction pathway, in the vitreous of VRL patients.

### 3.6. Comparative Analysis of Selected miRNA Gene Targets and Differentially Expressed Genes in VRL

To better define the role played by miRNAs in the pathogenesis of VRL, we sought to select miRNAs that are able to target genes modulated in VRL. Therefore, we used a more sophisticated integrative database for human microRNA target prediction (mirDB: http://www.mirdb.org/) to obtain a list of genes that are targeted by each of the selected miRNAs, with very high scores of over 95 (fold change > 7). DAVID analysis suggested the relationship of the miRNAs with pathways as shown in Appendix A.

This analysis found no significant relationship with ECM-receptor interaction, but detected laminin gamma (LAMC) 3 related to ligands that bind to CD44. Thus, we were able to identify miRNAs that may regulate gene modulation involved in the pathogenesis of VRL. Appendix A shows the targeted genes and their corresponding targeting miRNAs.

### 3.7. Vitreous IL-10 Levels and miRNAs

Because elevated vitreous IL-10 level is essential to establish a diagnosis of VRL, we examined the correlation between vitreous IL-10 levels and the expression of 30 miRNAs (Figure 9).

Using the miRDB database, we identified the miRNAs that target IL10. Among them, we used Cytoscape platform to show miRNAs that were upregulated or downregulated in this analysis. This analysis revealed that miR-1236-3p correlated significantly and positively with vitreous IL-10 levels in VRL patients (Figure 10).

### 3.8. Association between Clinical Features and miRNAs

VRL patients were divided into two groups according to clinical features: diffuse vitreous opacities with or without multiple subretinal white lesions. We evaluated the association between vitreous miRNA expression and clinical features (Figure 11).

The vitreous expression levels of 4 miRNAs (miR3677-3p, miR-423-3p, miR-548ay-3p, miR-520h) were significantly (*p* < 0.05) elevated in VRL manifesting diffuse vitreous opacity with multiple subretinal white lesions than in VRL with diffuse vitreous opacity alone. On the other hand vitreous expression levels of 8 miRNAs (miR-107, miR-1269b, miR-4461, miR-7706, miR-6503-5p, miR-526b-5p, miR-7848-3p, miR-4422) were significantly (*p* < 0.05) elevated in VRL manifesting diffuse vitreous opacities alone than in VRL with vitreous opacities and multiple subretinal white lesions.

### 3.9. Machine Learning and miRNA Expression Validation in VRL

Machine learning analysis using random forest algorithm revealed that among the 17 differentially expressed miRNAs (miR-361-3p, miR-3927-5p, miR-3941, miR-6817-5p, miR-4537, miR-4724-5p, miR-338-5p, miR-6799-3p, miR-550b-2-5p, miR-196a-5p, miR-8084, miR-589-3p, miR-708-5p, miR-5692b, let-7c-5p, miR-566, miR-4306) observed in the study, vitreous miR-361-3p was the best predictor for VRL; when the cutoff value was 0.480, the maximum accuracy was 0.875, and area under the ROC curve was 0.921. ROC Pathway analysis using these 17 microRNAs also suggested 4 microRNAs related to ECM receptors (Table 4).

## 4. Discussion

In the present study, we assessed the expression of 2565 miRNAs in both vitreous and serum samples obtained from VRL patients, disease controls and healthy controls, and identified miRNAs that are biomarkers of VRL, defined as the area under the ROC curve > 0.8 for VRL versus uveitis with vitreous opacity. An area under the ROC curve of 0.8 is considered moderately accurate for prediction. High throughput miRNA analysis provides a huge number of miRNAs to facilitate comprehensive analysis of disease status, and to elucidate miRNA profiles and pathways that distinguish VRL from chronic uveitis with vitreous opacity and from healthy status. Identification of vitreous and serum miRNA alterations may improve current diagnostic methods for VRL by substantiating the evidence of disease pathogenesis. Interestingly, among all the differentially expressed miRNAs observed in this study, the most prominent putative biomarker of VRL was miR-361-3p, as calculated by random forest. The ROC curve analysis of miR-361-3p yielded area under the curve of 0.921 (95% CI 0.806-1) for discriminating VRL from vitreous opacity with uveitis, indicating diagnostic value. miR-361-3p has been reported to be modulated in solid tumors such as colon, lung, pancreas, cervical, thyroid, cervical cancer and retinoblastoma [26,27,28,29,30,31]. These findings allow us to speculate that increased vitreous miR-361-3p may contribute to carcinogenesis in VRL patients, and the performance of vitreous miR-361-3p needs further validation before clinical use.

In this analysis, we found 1478 dysregulated miRNAs in vitreous samples of VRL patients compared with patients with macular hole/epiretinal membrane. As the number of differentially expressed vitreous miRNAs is relatively high, we focused on miRNAs with more than seven-fold changes in expression (miR-30b-3p, miR-1290 and miR-21-5p; Appendix A). The majority of data about these miRNAs come from studies focused on various types of cancers, regarding their roles in cell proliferation, apoptosis, division, migration and invasion. Some of the studies analyzed hepatocellular carcinoma, non-small lung carcinoma, and esophageal squamous cell carcinoma [32,33,34,35]. The cellular infiltrate present in the vitreous of VRL patients suggests that lymphoma-derived and infiltrated leukocyte-derived miRNAs may contribute to the vitreous miRNA profile obtained in this study. However, resident cells may produce other miRNAs. Therefore, the differentially expressed miRNAs we identified in this study may better reflect the total miRNAs, rather than miRNA production from only the lymphoma and infiltrating leukocytes.

Early diagnosis of VRL is often challenged by the absence of accurate early diagnostic and prognostic biomarkers, because VRL may masquerade as chronic uveitis. Although vitreous IL-10 analysis had the highest diagnostic sensitivity [5,36], serum IL-10 is not detected in all patients with VRL. Effective primary screening using peripheral blood has not been reported. To the best of our knowledge, this is the first report of the potential of using circulating miRNAs as candidate biomarkers to identify VRL patients, especially those with vitreous opacity. Furthermore, the strong point of our study is that we studied a larger number of miRNAs (2565 miRNAs) in VRL patients compared to previous reports [16,17], which allows reliable statistical analysis. A lack of predictive serum biomarker is the main reason for delayed diagnosis of VRL. In the absence of unique cytological or laboratory features in peripheral blood which may facilitate diagnosis, diagnostic vitrectomy is currently used to define and diagnose VRL. However, diagnostic vitrectomy potentially carries the risk of complications. Therefore, less invasive tools to differentiate VRL from vitritis are needed. Recent studies have found that miRNAs are present in plasma in detectable levels. Due to their small size and stem-loop structure, they are more stable than messenger RNAs in body fluids, are resistant to degradation, and are easily and rapidly measurable [37,38,39,40]. Compared to diagnostic vitrectomy, blood-based biomarker assays are relatively economical and non-invasive methods to detect VRL, with additional advantages of easy accessibility and low risk associated with sample collection. Thus, we assessed the serum miRNA expression levels and found that the levels of ten miRNAs (mir-1273d, 133b, 146a-5p, 181a-5p, 193b-3p, 221-3p, 326, 345-5p, 422a, 4655-3p) were different between VRL and macular hole/epiretinal membrane patients, whereas the levels of three miRNAs (miR-6513-3p, 4445-3p, 138-2-3p) were higher in VRL than in uveitis with vitreous opacity. Among the differentially expressed miRNAs in vitreous obtained by comparing VRL versus uveitis with vitreous opacity, three modulated (upregulated) vitreous miRNAs (miR-6513-3p, 138-2-3p and 4445-3p) may be used for distinguishing VRL from uveitis with vitreous opacity, since principal component analysis of these miRNAs clearly separated patients in two distinct groups, as shown in Figure 6. In our study, ROC curve analysis showed that serum miR-326 expression level could discriminate VRL from macular hole/epiretinal membrane, miR-6513-3p could discriminate VRL from uveitis with vitreous opacities, and a combination of these two miRNAs may have even higher performance. Further evaluation is needed to confirm these very preliminary results. Clinically, discriminating between VRL and uveitis with vitreous opacity using serum miRNAs may support clinical decision making and allow timely initiation of treatment, thereby avoiding vitrectomy.

Previous studies have investigated miRNA profiles in diffuse large B-cell lymphoma (DLBCL) [41,42,43,44,45,46]. miR-181a-5p is reported to be upregulated in peripheral blood of acute lymphoblastic leukemia compared with normal peripheral blood mononuclear cells [47]. miR-422a has been associated with better overall survival in patients with DLBCL, and is related to patient’s response to chemotherapy [48]. MiRNA-133b is one of the most widely researched RNAs and has been implicated in many cancers such as ovary, lung, liver and esophageal cancers [49,50,51,52]. miR-146a-5p is a tumor suppressor in malignancies, regulating proliferation and apoptosis. Its role in inflammation is less well studied but has been suggested to be a potential biomarker for rheumatoid arthritis [53,54,55]. Furthermore, similar to rheumatoid arthritis patients, miR-146a-5p expression levels are reduced in hepatocellular cell carcinoma [56]. miR-146a is involved in B cell hyperplasia [57], and miR-146a-5p has been reported to be a useful prognostic biomarker for DLBCL [58]. The possible roles of miR-326 as tumor suppressor miRNA have been established in various carcinomas. Given the above background, we investigated the possible pathways by which miRNA may be involved in VRL. According to KEGG analyses, the functions of most pathways and genes related to VRL are linked to Huntington’s disease, circadian entrainment, ECM-receptor interaction, antigen processing and presentation, and morphine addiction pathways. These findings suggest that the pathogenesis of VRL may be related to disruption of these biological pathways. The above signaling pathways, particularly ECM-receptor interaction pathway, are closely related to the pathogenesis of VRL. The ECM‒receptor interaction pathway plays a crucial role in breast cancer and kidney cancer [25,59], and miRNAs have been shown to regulate this pathway [60]. The KEGG pathway analysis showed substantial overlap among the signal transduction pathways of target gene sets involved in cancer transcriptional dysregulation pathway, proteoglycan, and cancer pathways, suggesting that these pathways may be closely related to cancers including VRL. CD44 expression is upregulated in DLBCL [61,62,63]. In addition, there is a report that CD44 expression is upregulated via ECM-receptor interaction in kidney cancer [59]. Thus, ECM receptor interaction may be an important pathway also in VRL.

There are some discrepancies regarding miR-92, miR-19b, and miR-21. Kakkassery et al. [16]. reported that these three miRNAs were significantly upregulated in VRL patients. On the contrary, another study that assayed 168 miRNAs reported that these three miRNAs were not differentially expressed [17]. Our results of increased miR-21 expression in VRL patients are in concordance with the report by Kakkassery et al. [16]. Especially, miR-21 plays a pivotal role in the regulation of master regulator transcription factor essential in B-cell biology [64].

Pathways related to upregulated or downregulated miRNAs were mostly related to proteoglycans in cancers, possibly due to its post-transcriptional regulation by miRNAs. Proteoglycans are well known to be involved in DLBLC, contributing to tumor growth and metastasis [61,62,63]. Our findings of the relevance of proteoglycans in DLBLC are consistent with early studies by Nagel et al. [65], who reported a prognostic gene expression signature in DLBLC tumor tissue, which included the proteoglycan component. Hippo signaling is also an important network involved in the regulation of cancer cells and has been reported to be dysregulated in hematological malignancies including myelogenous leukemia [66,67,68]. A wide variety of inflammatory conditions strongly suggest that these miRNAs function in immune‒mediated pathways. Pathway enrichment analysis for miRNA target genes supports the hypothesis that these miRNAs regulate inflammatory and interleukin signaling, consistent with inflammatory conditions in VRL [69].

The genes targeted by seven upregulated miRNAs (fold change > 7) in VRL are primarily related to pathways in cancers. Several lines of evidence support the role of modulated miRNAs in the pathogenesis of DLBCL. According to the miRDB database (http://www.mirdb.org/), downregulation of vitreous miR-4795-3p and miR-29b-2-5p may be related to the reported increase of their target gene MYD88, which is strongly associated with VRL pathogenesis [70,71]. The induction of miRNAs related to signaling in VRL is an interesting finding. According to the canonical miRNA‒mRNA interaction, upregulation of these posttranscriptional elements in VRL may promote downregulation of important genes responsible for B-cell lymphoma. In fact, 21-5p has been associated with survival and proliferation of malignant cells in DLBCL [43,72].

We could not find any literature on the roles of miR-4445-3p in carcinogenesis and the immune system. Expression levels of this miRNA in both vitreous and serum were higher in VRL patients than in controls, suggesting that this miRNA is oncogenic. High expression of this miRNA in vitreous and serum could provide insight into its role, although further study is needed to analyze the its function. Since only very few miRNAs and their target genes have been identified, and the functions of most miRNAs remain unclear [73], such insight into the interplay of miRNAs in VRL would help further steer rational diagnostic and therapeutic strategies and ultimately improve patient outcomes.

MiR-1236-3p is significantly downregulated in gastric cancer and lung cancer relative to healthy persons [74,75], although to our knowledge its function remains unknown. Vitreous IL-10 strongly correlates with miR-1236-3p expression. IL-10 is an inhibitory cytokine that regulates tumor immune responses. In our series, all patients with VRL showed serum IL-10 levels below detection limit. MiR-1236-3p has been shown to regulate the PI3K/Akt signaling pathway [74], which is important for B-cell proliferation and differentiation.

Currently, it remains unclear why VRL mainly manifests diffuse vitreous opacity and multiple retinal or subretinal white lesions. An interesting finding of our study is that significantly different microRNA expression patterns exist between vitreous opacity with subretinal infiltration and vitreous opacity alone in patients with VRL, suggesting that miRNAs as regulators of molecular pathways impact the phenotypes of VRL. Lymphoma cells, infiltrating leukocytes, and ocular resident cells such as retinal pigmented epithelium and glial cells can produce miRNAs [76,77,78,79]. Therefore, the results presented here most likely represent the combined miRNA expression from lymphoma cells, leukocytes, and ocular resident cells, possibly interacting with each other. This study may provide valuable insight into the pathogenesis of each phenotype of VRL.

The present study had several limitations. First, the study was carried out using retrospectively collected samples. Consequently, sample handling conditions before microarray analysis, such as the interval between centrifugation and storage and the storage temperature, were not strictly controlled. Although miRNAs are more stable than messenger RNA, various processes can affect their levels in serum [38,39]. Further prospective studies are required to confirm our findings. Second, due to the relatively small sample size because VRL is a rare disease, the statistical significance is limited. Third, this investigation was a single-center case-control study, which does not allow calculation of positive and negative predictive values. However, because of the rarity of VRL and vitritis, conducting prospective cohort studies is difficult in terms of cost and time. Fourth, the large number of female patients and multiple forms of uveitis included in the control group may introduce some variability and confounders to the control group, which may prevent identification of other lymphoma-specific miRNA expression. Further study is needed to validate the present findings by including more patients with VRL compared with a carefully matched control group.

## 5. Conclusions

This work represents the first analysis performed on a large number of miRNAs integrated with study of the profiles of gene expression in VRL. Using this approach, we are able to identify the specific molecular pathways in which regulation by these miRNAs may occur. Further studies are needed to confirm the present observations and to clarify whether expression of the miRNAs is related to extraocular involvement and poor prognosis. Nonetheless, this study sheds light on some epigenetic aspects of VRL by identifying specific miRNAs, which may represent promising candidates for the identification of disease biomarkers and the design of novel therapeutic strategies in VRL.

## Figures and Tables

**Figure 1 jcm-09-01844-f001:**
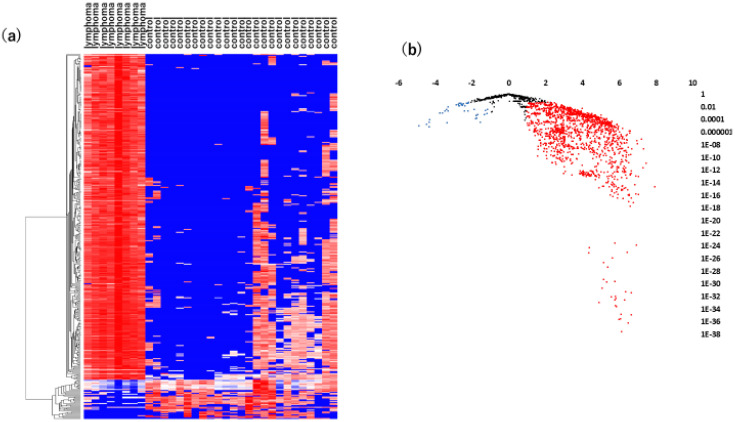
Analyses of vitreous miRNAs in VRL versus heathy controls. (**a**) Unsupervised hierarchical clustering analysis with a heatmap. The spectrum of dark red to dark blue color corresponds to high to low values. (**b**) Volcano plot of the fold change of miRNAs. Blue dots indicate downregulation, red dots indicate upregulation of miRNA and black dots indicate no significant difference.

**Figure 2 jcm-09-01844-f002:**
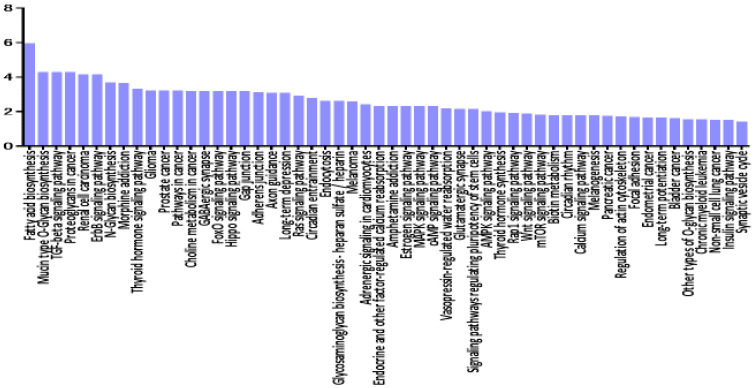
Histogram presenting pathways enriched in genes targeted by VRL-modulated miRNAs and differentially expressed miRNAs in vitreous of VRL patients compared with macular hole/epiretinal membrane patients. y axis: −log_10_ (*p*-value). TGF: transforming growth factor, ECM: extracellular matrix, FoxO: forkhead box O, Rap: rhoptry-associated protein, AMPK: AMP-activated protein kinase, PI3K-Akt: phosphatidylinositol 3-kinase- protein kinase B, MAPK: mitogen-activated protein kinases, mTOR: mechanistic target of rapamycin, cGMP-PKG: cyclic guanosine monophosphate -dependent protein kinase, HIF-1: hypoxia-inducible factor-1, HTLV-1: human T-lymphotropic virus type-1.

**Figure 3 jcm-09-01844-f003:**
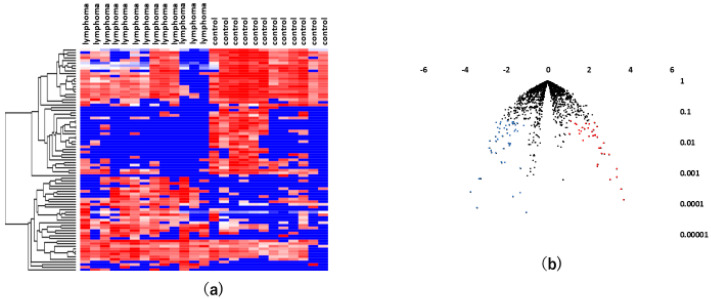
Analyses of serum miRNAs in VRL versus heathy controls. (**a**) Unsupervised hierarchical clustering analysis with a heatmap. The spectrum of dark red to dark blue color corresponds to high to low values. (**b**) Volcano plot of the fold change of miRNAs. Blue dots indicate downregulation, red dots indicate upregulation of miRNA and black dots indicate no significant difference.

**Figure 4 jcm-09-01844-f004:**
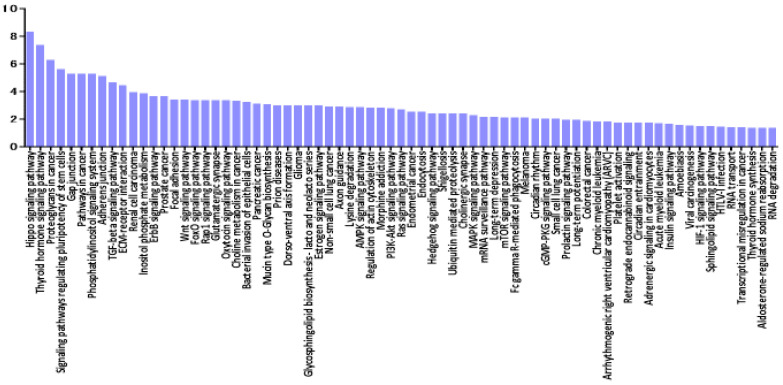
Histogram presenting pathways enriched in genes targeted by VRL-modulated miRNAs and differentially expressed miRNAs in serum of VRL patients compared with macular hole/epiretinal membrane patients. y axis: −log_10_ (*p*-value). TGF: transforming growth factor, GABA: gamma-aminobutyric acid, FoxO: forkhead box O, MAPK: mitogen-activated protein kinases, cAMP: cyclic adenosine monophosphate, AMPK: AMP-activated protein kinase, Rap: rhoptry-associated protein, mTOR: mechanistic target of rapamycin.

**Figure 5 jcm-09-01844-f005:**
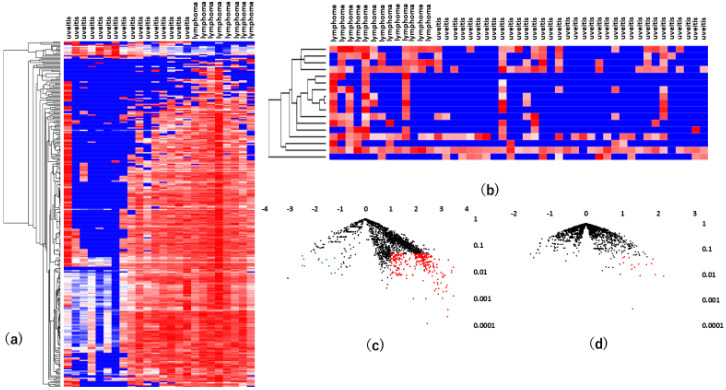
Results of analyses of differentially expressed vitreous and serum miRNAs in VRL patients versus uveitis with vitreous opacity. (**a**). vitreous, (**b**). serum unsupervised hierarchical clustering analysis with a heatmap. (**c**). vitreous, (**d**). serum volcano plot showing differentially expressed miRNAs in VRL versus uveitis. Blue dots indicate downregulation, red dots indicate upregulation of the miRNAs and black dots indicate no significant difference.

**Figure 6 jcm-09-01844-f006:**
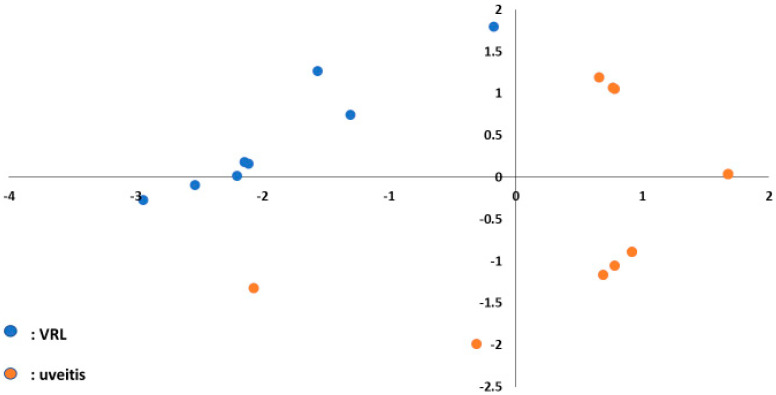
Principal component analysis plot to separate VRL and uveitis using three upregulated miRNAs (miR-6513-3p, 138-2-3p and 4445-3p) in the vitreous. Blue dots indicate VRL and orange dots indicate uveitis with vitreous opacity.

**Figure 7 jcm-09-01844-f007:**
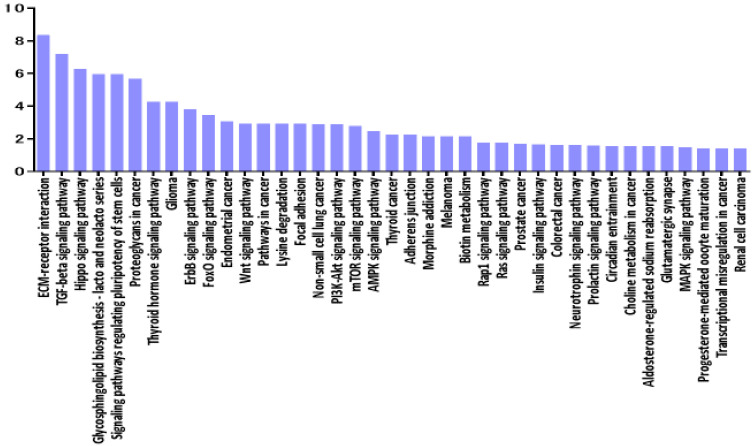
Histogram presenting pathways enriched in differentially expressed mRNAs in vitreous of VRL patients compared with uveitis with vitreous opacity. y axis: −log_10_ (*p*-value). ECM: extracellular matrix, TGF: transforming growth factor, FoxO: forkhead box O, PI3K-Akt: phosphatidylinositol 3-kinase- protein kinase B, mTOR: mechanistic target of rapamycin, AMPK: AMP-activated protein kinase, Rap: rhoptry-associated protein, MAPK: mitogen-activated protein kinases.

**Figure 8 jcm-09-01844-f008:**
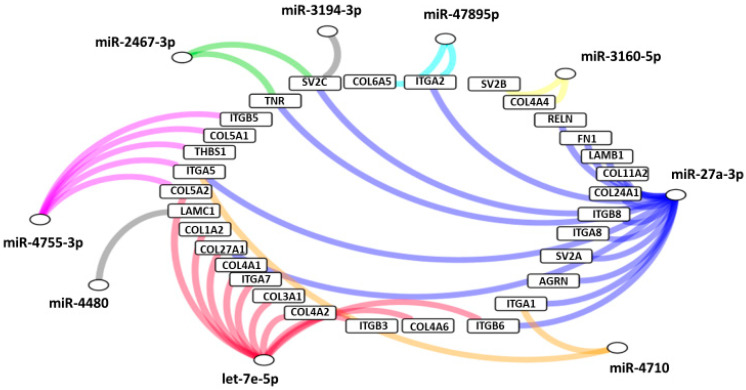
Predicted genes targeted by miRNAs in extracellular matrix receptor interaction pathway, in the vitreous of VRL patients compared with uveitis. Selected inflammatory genes targeted by each group of miRNAs are depicted in a network diagram. Differentially expressed miRNAs that do not target any of the selected genes are not shown. AGRN: agrin, COL: collagen, FN: fibronectin, ITGA; integrin alpha, ITGB: integrin beta, LAMB: laminin beta, LAMC: laminin gamma, RELN: reelin, SV: synaptic vesicle glycoprotein, THBS: thrombospondin, TNR: tenascin R.

**Figure 9 jcm-09-01844-f009:**
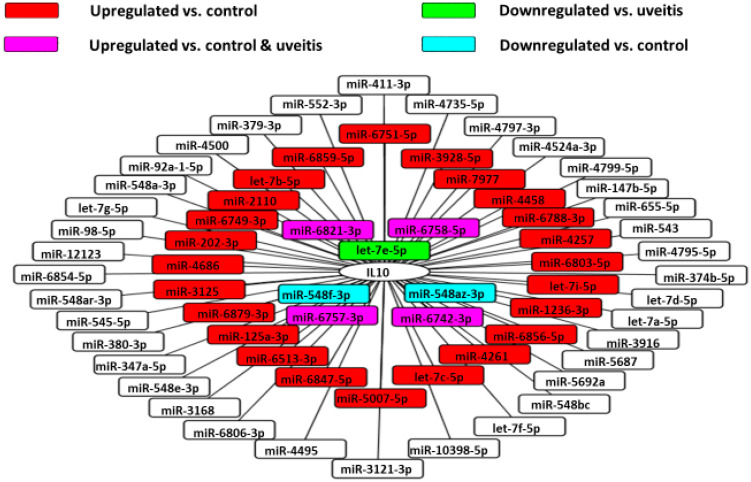
MiRNA‒IL-10 gene interaction networks in VRL. 67 validated microRNAs having interactions with IL-10 gene in the vitreous of VRL patients are shown.

**Figure 10 jcm-09-01844-f010:**
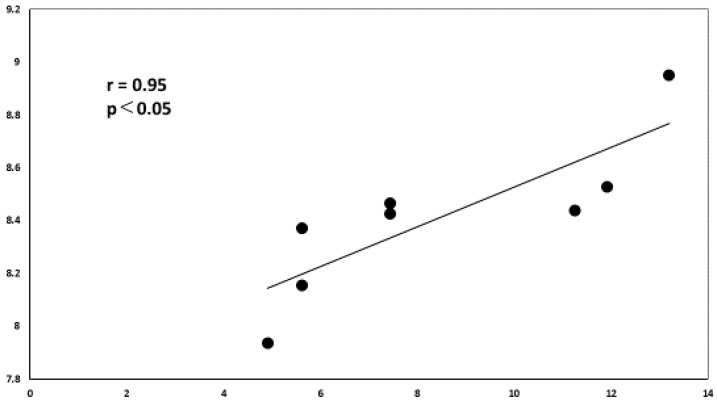
Correlation diagram between IL-10 and miR-1236-3p. There is a positive correlation between the relative expression levels of IL-10 and miR-1236-3p in the vitreous of VRL patients (*r* = 0.952, *p* < 0.05).

**Figure 11 jcm-09-01844-f011:**
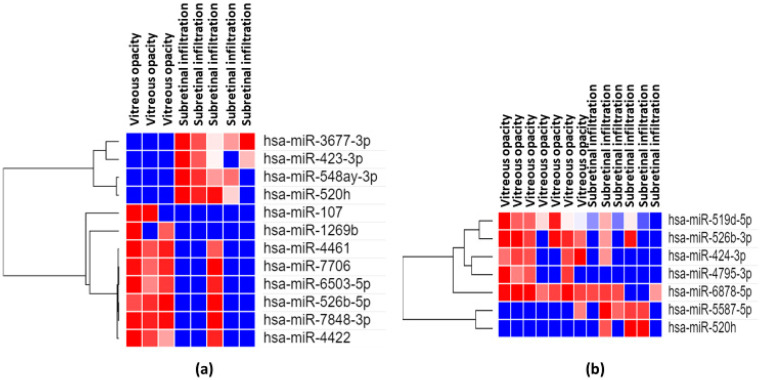
Results of analyses of differentially expressed miRNAs for VRL manifesting vitreous opacity alone vs. vitreous opacity with subretinal infiltration. Heatmaps depict expression profiles of mRNAs in vitreous (**a**) and serum (**b**) The spectrum of dark red to dark blue color corresponds to high to low values.

**Table 1 jcm-09-01844-t001:** Demographic, clinical and laboratory data of vitreoretinal lymphoma (VRL) patients and controls.

			Vitreoretinal Lymphoma	Uveitis with Vitreous Opacity	Macular Hole/Epiretinal Membrane	Healthy Individuals
N			14	40	26	12
Age			66.6 ± 12.0	68.1 ± 14.1	68.7 ± 9.5	53.9 ± 20.5
Sex	Male		5	8	12	6
	Female		9	32	14	6
Clinical features	Main ocular lesion: vitreous		7	40	NA	NA
	sub-retina		7	0	NA	NA
	Primary organ: eye		7	NA	NA	NA
	Brain involvement		10	NA	NA	NA
	Cytopathology (>class IV)		1	NA	NA	NA
	IgH positivity		11	NA	NA	NA
	IL-10/IL-6 ratio (>1)		14	NA	NA	NA
	White blood cell (/μL)		6392	6626	5952	6562
	IL-2 receptor (U/mL)		323	416	NA	NA

NA: not available.

**Table 2 jcm-09-01844-t002:** Vitreous miRNAs modulated in vitreoretinal lymphoma versus control and uveitis with vitreous opacity, showing the top 20 in descending order of *p* value.

vs. Control; Up	vs. Uveitis; Up	vs. Control; Down	vs. Uveitis: Down
miR-629-3p	miR-3677-3p	miR-519c-5p, miR-523-5p miR-518e-5p, miR-522-5p miR-519a-5p, miR-519b-5p	miR-519c-5p, miR-523-5p miR-518e-5p, miR-522-5p miR-519a-5p, miR-519b-5p
miR-6740-5p	miR-6513-3p	miR-106a-3p	miR-27a-3p
miR-4647	miR-5094	miR-516a-5p	miR-3160-5p
miR-6780a-3p	miR-152-5p	miR-6869-3p	miR-2467-3p
miR-657	miR-892a	miR-548s	miR-4710
miR-1266-3p	miR-300	miR-614	miR-4718
miR-4714-5p	miR-433-3p	miR-548f-3p	let-7e-5p
miR-3160-3p	miR-4650-5p	miR-4718	miR-504-5p
miR-605-5p	miR-3927-5p	miR-3194-3p	miR-6070
miR-4330	miR-6841-5p	miR-3160-5p	miR-4755-3p
miR-7155-3p	miR-3973	miR-4480	miR-3194-3p
miR-3120-5p	miR-4524a-5p	miR-2467-3p	miR-191-5p
miR-216b-3p	miR-654-3p	miR-29b-2-5p	miR-526a, miR-520c-5p miR-518d-5p
miR-4756-3p	miR-6505-3p	miR-29b-3p	miR-4480
miR-4733-3p	miR-4756-3p	miR-646	miR-4789-5p
miR-6716-3p	miR-146b-3p	miR-515-5p	
miR-1184	miR-603	miR-6070	
miR-8070	miR-3126-3p	miR-4755-3p	
miR-4692	miR-5194	miR-31-3p	
miR-30b-5p	miR-7973	miR-6835-3p	

Up: upregulate, Down: downregulate.

**Table 3 jcm-09-01844-t003:** Vitreous and serum miRNAs modulated in vitreoretinal lymphoma versus uveitis with vitreous opacity [for uveitis up (vitreous) - the top 20 are shown] (*p* < 0.05).

vs. Uveitis Up (Vit)	vs. Uveitis Up (Serum)	vs. Uveitis Down (Vit)	vs. Uveitis Down (Serum)
miR-3677-3p	miR-4475	miR-519c-5p, miR-523-5p miR-518e-5p, miR-522-5p miR-519a-5p, miR-519b-5p	miR-548p
miR-6513-3p	miR-27b-3p	miR-27a-3p	
miR-5094	miR-4684-5p	miR-3160-5p	
miR-152-5p	miR-1286	miR-2467-3p	
miR-892a	miR-7848-3p	miR-4710	
miR-300	miR-6500-5p	miR-4718	
miR-433-3p	miR-6513-3p	let-7e-5p	
miR-4650-5p	miR-5684	miR-504-5p	
miR-3927-5p	miR-369-5p	miR-6070	
miR-6841-5p	miR-500a-3p	miR-4755-3p	
miR-3973	miR-4445-3p	miR-3194-3p	
miR-4524a-5p	miR-4439	miR-191-5p	
miR-654-3p	miR-4434	miR-526a, miR-520c-5p miR-518d-5p	
miR-6505-3p	miR-138-2-3p	miR-4480	
miR-4756-3p	miR-181c-5p	miR-4789-5p	
miR-146b-3p			
miR-603			
miR-3126-3p			
miR-5194			
miR-7973			

Up: upregulate, Down: downregulate, Vit: vitreous.

**Table 4 jcm-09-01844-t004:** miRNAs related to ECM-receptor interaction among 17 miRNAs selected by random forest and their target genes.

microRNA	*p*-Value	Target Genes
miR-6817-5p	6.17E-12	COL6A5, THBS1, COLA6A, ITGA2, LAMC1, FN1
miR-6799-3p	0.02	COL6A6, COL4A1
miR-196a-5p	4.98E-07	COL24A1, COL3A1, COL1A2
let-7c-5p	7.78E-19	COL27A1, ITGB6, COL3A1, COL1A1, COL1A2, ITGA7, COL4A6, COL5A2, COL4A1

COL: collagen type, THBS: thrombospondin ITGA: integrin alpha, ITGB: integrin beta, LAMC: laminin gamma, FN: fibronectin.

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
