# Peer review of "High-Throughput MicroRNA Profiling of Vitreoretinal Lymphoma: Vitreous and Serum MicroRNA Profiles Distinct from Uveitis"

_jcm, 2020, doi:10.3390/jcm9061844_

Round 1

Reviewer 1 Report

This manuscript is well-written and the study is well-designed. The correlation between miRNAs expression and the diagnosis of VRL is confirmed systemically and scientifically. The experiments are performed correctly and the results are presented well without identifiable errors or significant bias. I think this modality can become a non-invasive diagnostic tool for VRL. It also generate several unanswered questions and hypotheses that can help guiding follow-up studies in the future.

Small comments below:

In the "Introduction", PCR analysis for IgH receptor rearrangement with detection of 75-80% is not low as reported by the authors. I prefer to refer to it as "suboptimal".

In the "Introduction", IL-10/IL-6 ratio was referred to by the authors but its value in diagnosing VRL was not explained. It should be explained.

In the "Methods", the table heading needs to be consistent with no words should be divided between lines

In the "Methods", in figures 1a, 3a, 5a and 5b the writing is too small and difficult to read.

Great work and excellent manuscript. I recommend proceeding with publication

Author Response

We are grateful to the reviewer for the encouraging comments of our work. We also appreciate all the constrictive comments. Our responses to the comments are as follows.

  1. In the "Introduction", PCR analysis for IgH receptor rearrangement with detection of 75-80% is not low as reported by the authors. I prefer to refer to it as "suboptimal".

RESPONSE: We agree with the reviewer’s suggestion. We have changed “low detection rates” to “suboptimal detection rates” (line 43).

  1. In the "Introduction", IL-10/IL-6 ratio was referred to by the authors but its value in diagnosing VRL was not explained. It should be explained.

RESPONSE: We thank the reviewer for the pertinent comment. We have added the diagnostic value of IL-10/IL-6 ratio as follows:

“On the other hand, IL-10/ IL-6 ratio is a useful diagnostic tool: a ratio higher than one has been reported to indicate VRL and a ratio lower than one uveitis [7]. However, our group has found that 10 to 20% of VRL cases have IL-10/IL-6 ratio less than 1.” (lines 47-50)

  1. In the "Methods", the table heading needs to be consistent with no words should be divided between lines

RESPONSE: We have improved the headings of Figure 1 and made sure that the headings of all figures are consistent.

  1. In the "Methods", in figures 1a, 3a, 5a and 5b the writing is too small and difficult to read.

RESPONSE: We have enlarged the fonts of the words in Figures 1a, 3a, 5a and 5b to make them legible.

Reviewer 2 Report

In their paper “High-throughput microRNA profiling of vitreoretinal lymphoma: vitreous and serum microRNA profiles distinct from uveitis,” Minezaki et al. describe the use of microRNA analysis of serum and vitreous samples from patients with primary vitreoretinal lymphoma, uveitis, and other controls to demonstrate several microRNAs that were differentially expressed in patients with primary vitreoretinal lymphoma, and also showed that miR-6513-3p levels in the serum were different between patients with primary vitreoretinal lymphoma and other disease entities. This paper presents an important and novel finding, as the identification of serum markers and additional vitreous markers would aid significantly in the ability to accurately diagnose primary vitreoretinal and primary CNS lymphoma.

In table 1 it would be helpful to include the N for number of total patients in each group at the top of the column for clarity. The text for the rows in the table also appears to be mis-aligned and it would help to reformat to make the table easier to read.

There are few grammatical errors throughout the paper that should be corrected for ease of reading. For example, the opening line of the abstract might more clearly read “Vitreoretinal lymphoma (VRL) is a non-Hodgkin lymphoma of the diffuse large B cell type (DLBCL), which is an aggressive cancer causing central nervous system-related morbidity and mortality.” Line 56 should read “that have the function of suppressing gene translation…” etc.

Could the authors comment as to whether the primary vitreoretinal lymphoma patients with different serum levels of miR-6513-3p did or did not have CNS involvement? It would be useful to know if this biomarker could play a role in identifying those patients with CNS disease.

Author Response

We are grateful to the reviewer for the encouraging comments of our work. We also appreciate all the constrictive comments. Our responses to the comments are as follows.

  1. In table 1 it would be helpful to include the N for number of total patients in each group at the top of the column for clarity. The text for the rows in the table also appears to be mis-aligned and it would help to reformat to make the table easier to read.

RESPONSE: We thank the reviewer for the suggestion. We have added N for number of total patients in the heading of Table 1. We have also reformatted the table to make it easier to read.

  1. There are few grammatical errors throughout the paper that should be corrected for ease of reading. For example, the opening line of the abstract might more clearly read “Vitreoretinal lymphoma (VRL) is a non-Hodgkin lymphoma of the diffuse large B cell type (DLBCL), which is an aggressive cancer causing central nervous system-related morbidity and mortality.” Line 56 should read “that have the function of suppressing gene translation…” etc.

RESPONSE: We apologize for the grammatical errors. We have corrected those kindly pointed out by the reviewer and checked the whole manuscript thoroughly for further errors.

  1. Could the authors comment as to whether the primary vitreoretinal lymphoma patients with different serum levels of miR-6513-3p did or did not have CNS involvement? It would be useful to know if this biomarker could play a role in identifying those patients with CNS disease.

RESPONSE: We found that in patients with primary vitreoretinal lymphoma, those with high miR-6513-3p expression had apparently fewer CNS involvement, but the difference did not reach statistical significance (p = 0.43).